# Structural Characterization and Adsorption Capability of Carbonaceous Matters Extracted from Carbonaceous Gold Concentrate

**Huiqun Niu, Hongying Yang * and Linlin Tong**

Key Laboratory for Ecological Metallurgy of Multi-Metallic Mineral (Ministry of Education), School of Metallurgy, Northeastern University, Shenyang 110819, China; niuhuiqun525@126.com (H.N.); tongll@smm.neu.edu.cn (L.T.)
* Correspondence: yanghy@smm.neu.edu.cn; Tel.: +86-138-8980-3669

**Abstract:** In this paper, the structures of element carbon and humic acid extracted from carbonaceous gold concentrate were characterized employing a variety of analytical methods. The extracted amounts of ECE (elemental carbon extract) and HAE (humic acid extract) were 14.84–38.50 and 11.55–28.05 mg g$^{-1}$, respectively. SEM and porosity analysis indicated that ECE occurred mostly as irregular blocky particles with a mesoporous surface with the average pore diameter being 31.42 nm. The particle size of ECE was mainly ranged from 5.5 to 42 μm and the specific surface area was 20.35 m$^2$ g$^{-1}$. The physicochemical features and structure of ECE were close to activated carbon, and the crystallinity was slightly lower than graphite. The particle size distribution of HAE varied from 40 to 400 nm with the specific surface area of 42.84 m$^2$ g$^{-1}$, whereas the average pore diameter of HAE was 2.97 nm. FTIR and UV–VIS analyses indicated that HAE was a complex organic compound containing the enrichment of oxygen-containing structure. The results showed that the adsorption amounts of ECE and HAE under the acidic conditions were 470.46 and 357.60 mg g$^{-1}$, respectively. In an alkaline environment, the amount of ECE was 449.02 mg g$^{-1}$ and the value of HAE was 294.72 mg g$^{-1}$. ECE mainly utilized the outer surface and mesoporous structure to adsorb gold, while the functional groups' complexation or surface site adsorption was the leading approach for HAE to adsorb gold.

**Keywords:** carbonaceous gold concentrate; carbonaceous matters; elemental carbon; humic acid; structural characterization; adsorptive capacity

## 1. Introduction

The definition of carbonaceous gold ore is a refractory ore containing carbonaceous matters, and the carbonaceous matters in the ore can interact with gold–cyanide complexes. The presence of carbonaceous matters in carbonaceous gold ores will extremely interfere with the gold leaching process, preventing solution access and causing a decrease in leaching rate of gold. In the process of gold leaching, the adsorption/complexation between carbonaceous matters and gold–cyanide complexes is called the "preg-robbing" phenomenon. Especially, studies have shown that the carbonaceous matters can trigger the "preg-robbing" phenomenon when the content in a gold ore exceeds 0.2% [1]. In addition, differences in the structure of carbonaceous matters can lead to different adsorption capacities. Thus, an in-depth study of the structural composition of carbonaceous matters is of great significance in characterizing its "preg-robbing" capability. The main carbonaceous matters in a carbon-bearing gold ore can be classified into inorganic carbon and organic carbon. Of which, inorganic carbon is mainly composed of elemental carbon such as activated carbon and graphite, while organic carbon is mainly divided into humic substances and polymer hydrocarbons [2,3].

Elemental carbon can be found in almost all carbonaceous gold ores with a porous surface and high surface activity. The structure of elemental carbon is similar to that of

amorphous carbon and crystalline graphite [4]. Investigations by Adams have indicated that the presence of activated carbon will enhance the "preg-robbing" capability during cyanidation [5]. Besides, the "preg-robbing" capability of carbonaceous matters is closely related to the structural maturity of carbon. The higher the carbon maturity, the stronger the capacity of "preg-robbing" [3,6]. Moreover, research also indicated that carbonaceous matters with graphite structure possess the "preg-robbing" capability, and the chemical bonds corresponding to the graphite may play a significant role in the adsorption process [7]. Generally, surface adsorption is an outstanding way for elemental carbon to adsorb gold by utilizing porous structure and active sites. The adsorption behavior of elemental carbon is remarkably similar to that of activated carbon due to the structural similarity [3,6]. Therefore, the adsorption mechanism of elemental carbon in carbonaceous gold ore can be explained by the theory of activated carbon.

Humic acid obtained from different origins and geological conditions exhibits different molecular framework [8]. Humic acid is an important component of humic substances that is insoluble in acid but soluble in alkaline solution [9,10]. Some studies also inferred that humic acid is a complex macromolecular organic compound [11]. Analysis indicated that the chemical structure of humic acid is composed of aromatic rings or various heterocyclic rings as a basic skeleton with phenolic hydroxyl, carboxyl, and other oxygen-containing functional groups despite the different agglomerate shape [3,12,13]. Owing to its unique surface structure and chemical compositions, humic acid is more likely to produce strong adsorption or complexation for gold during the extracting process [14]. The generation of the gold-adsorbing performance in terms of humic acid can be simply classified into two points: first, humic acid possesses many active oxygen-containing functional groups, which can undergo complexation and ion exchange reactions to affect the gold extraction, and second, humic acid has an adsorptive potential for aurocyanide or other precious metals because of the sizeable specific surface area [15].

Structural analysis of carbonaceous matters separated from carbonaceous gold concentrate required a variety of analytical techniques to characterize and describe, which were carried out in this study, due to the particularity and structural complexity. The information on the structural characteristics of carbonaceous matters can be used to analyze and characterize the mechanism and capability of gold adsorption. Eventually, the chemical structure and composition of carbonaceous matters were revealed, and the gold-adsorbing efficiency resulted from the carbonaceous matters in gold concentrate was proved.

## 2. Materials and Methods

### 2.1. Sample

In this study, carbonaceous gold concentrate sample obtained from a gold deposit in Guizhou Province, China, was used for extracting ECE and HAE. The elementary composition of the sample is listed in Table 1. It can be seen that the ore sample contained Au 15.9 $g \cdot t^{-1}$, Fe 23.17%, S 24.7%, and As 3.41%. Carbon analysis showed that the contents of total carbon, inorganic carbon, and organic carbon were 6.06%, 1.70%, and 4.36%, respectively. XRD analysis of the gold concentrate indicated that the main metallic minerals were pyrite, arsenopyrite, and magnetite; quartz, muscovite, and dolomite were the main gangue minerals (Figure 1).

**Table 1.** The main element contents of carbonaceous gold concentrate (value in wt.%).

| Element | Au | Fe | S | As | Total Carbon | Inorganic Carbon | Organic Carbon |
|---|---|---|---|---|---|---|---|
| Content | 15.9 $g \cdot t^{-1}$ | 23.17 | 24.7 | 3.41 | 6.06 | 1.70 | 4.36 |

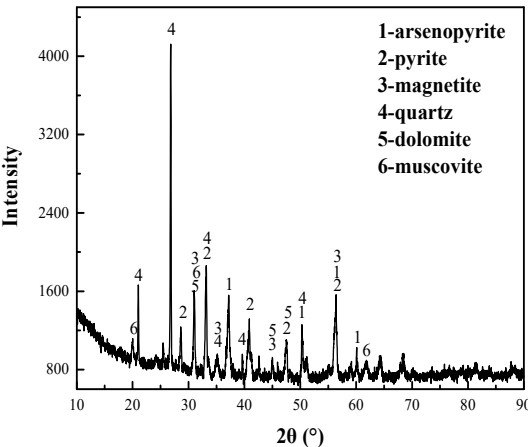

**Figure 1.** XRD diagram of carbonaceous gold concentrate.

## 2.2. Reagents

Activated carbon (analytical reagent) and graphite (analytical reagent) used in this article were provided from Dongxing Pharmaceutical Factory, Shenyang, China. Humic acid (HA) was purchased from Amresco, Solon, OH, USA.

## 2.3. Extraction of Carbonaceous Matters

### 2.3.1. Extraction of ECE

The extraction method of ECE is summarized according to literature review, which can be described as follows: (1) 4 M HCl was added into the cleaned concentrate at 60 °C to remove the carbonate minerals until all bubbles were disappeared and then centrifuged at 3000 r·min$^{-1}$ for 15 min; (2) the residue was added with 5.6 M HF at room temperature and agitated for 8 h and then centrifuged at 3000 r·min$^{-1}$ for 10 min. After that, the residue was rewashed with 1 M $H_2SO_4$; (3) NaOH solution and deionized water were, respectively, added into the residue to adjust the pH to neutral; (4) At last, $ZnCl_2$ solution was added, and the mixed solution was stirred for 20 min, then centrifuged for several times, and dried at 60 °C. The dark powder obtained was ECE, which was stored in vacuum for analysis.

### 2.3.2. Extraction of HAE

The extraction of humic substances is based on the method recommended by the International Humic Substances Society (IHSS). Humic substances were isolated using 1 M NaOH-$Na_4P_2O_7$ mixed solution as an extractant with a ratio of 4:1 (L: S) at 80 °C for 10 h. After that, the extract was stored at room temperature for 1 day and then centrifuged at 3500 r·min$^{-1}$ for 30 min. The extracting solution was acidified to pH 1–2 using HCl, then heated by a thermostatic water bath at 80 °C for 30 min, and preserved at 25 °C for 10 h. The supernatant and sediment fractions were separated by centrifuging at 2000 r·min$^{-1}$ for 10 min. The insoluble portion was added with 15 mL of ethanol to purify the precipitation and obtain the HAE. After that, HAE was rinsed with deionized water until almost neutral pH was achieved. Finally, the HAE was stored under vacuum condition.

## 2.4. Gold-Adsorbing Experiments

The gold-adsorbing experiments involving ECE and HAE were carried out in a solution containing 100 mg·L$^{-1}$ Au at 25 °C and 250 r·min$^{-1}$. The pH of the gold-containing solution was adjusted by NaOH or HCl. After 5 h of adsorption, the gold-containing solution was filtered, and the gold concentration in filtrates was assayed by atomic absorption spectroscopy (AAS). The adsorption amounts of ECE and HAE were calculated using Formula (1). Under the same conditions, gold-adsorbing experiments were also conducted

on activated carbon, graphite, and HA for comparison with ECE and HAE. The detailed experimental conditions of gold-adsorbing experiments are presented in Table 2.

$$q = (c_0 - c_t)\, V/M, \tag{1}$$

where M (g) is the mass of carbonaceous matters, $c_0$ (mg·L$^{-1}$) is the initial gold concentration, $c_t$ (mg·L$^{-1}$) is the gold concentration at the end of adsorption, V (mL) is the solution volume, and q (mg·g$^{-1}$) is the adsorption amount of gold.

**Table 2.** The detailed experimental conditions of gold-adsorbing experiments.

| Experimental Conditions | ECE | Activated Carbon | Graphite | HAE | HA |
|---|---|---|---|---|---|
| Initial Au concentration, mg·L$^{-1}$ | 100 | 100 | 100 | 100 | 100 |
| pH | 3, 12 | 3, 12 | 3, 12 | 3, 12 | 3, 12 |
| Temperature, °C | 25 | 25 | 25 | 25 | 25 |
| Rotating speed, r·min$^{-1}$ | 250 | 250 | 250 | 250 | 250 |
| Adsorption time, h | 5 | 5 | 5 | 5 | 5 |

### 2.5. Analytical Methods

Elemental analysis was conducted using a ONH analyzer (ONH836) and an infrared carbon-sulfur analyzer (CS230). The surface morphology of carbonaceous matters was characterized using a SEM (ULTRA PLUS). Pore structure and BET surface area were determined using an automatic physical adsorption analyzer (ASAP2020HD88) under nitrogen adsorption. FTIR spectra were tested using a Thermo Nicolet-380 spectrometer through pelleting the sample with 200 mg powder of KBr. UV–VIS spectra were provided on a TU-1901 UV–VIS spectrophotometer by recording adsorption spectra over the range of 200–800 nm. The X-ray diffraction was measured using Cu K$\alpha$ radiation at a scan speed of 0.5 °·min$^{-1}$ and within a range of $10° \le 2\theta \le 90°$. Raman spectra were measured in a HR800 Raman spectrometer using an argon laser ($\lambda$ = 663 nm) as an excitation source.

## 3. Result and Discussion

### 3.1. Quantification of Extracted ECE and HAE

The extraction amounts of ECE and HAE were calculated based on the proportion of total carbon in the gold concentrate using formula (2), and the amounts of extracted ECE and HAE accounted for per ton gold concentrate using formula (3). The results shown in Table 3 indicate that the extraction amounts of ECE and HAE are 14.84–38.50 and 11.55–28.05 mg·g$^{-1}$, respectively, or $0.90 \times 10^3$–$2.33 \times 10^3$ g·t$^{-1}$ and $0.79 \times 10^3$–$1.69 \times 10^3$ g·t$^{-1}$, respectively.

$$T = m_1 /(m_0 \times w_{carbon}) \tag{2}$$

$$G = m_1/m_0 \times 10^3, \tag{3}$$

where, $w_{carbon}$ is the percentage of total carbon in the gold concentrate, $m_1$ (mg) is the mass of ECE or HAE, $m_0$ (g) is the weight of concentrate used in the extraction process, T (mg·g$^{-1}$) is the amount of extracted ECE or HAE, and G (g·t$^{-1}$) is the content of ECE or HAE accounted for per ton of gold concentrate.

**Table 3.** The extracted amounts of elemental carbon extract (ECE) and humic acid extract (HAE).

| Carbonaceous Matters | Extraction Amount (mg·g$^{-1}$) | Content Accounted for per ton of Gold Concentrate (g·t$^{-1}$) |
|---|---|---|
| ECE | 14.84–38.50 | $0.90 \times 10^3$–$2.33 \times 10^3$ |
| HAE | 11.55–28.05 | $0.79 \times 10^3$–$1.67 \times 10^3$ |

*3.2. Structural Characterization of Carbonaceous Matters*

3.2.1. Elemental Analysis and Atomic Ratios

Table 4 presents the element's contents of carbonaceous matters. Carbon is the most important component accounting for 75.50% in ECE. Other elements include hydrogen (2.85%), oxygen (4.33%), and nitrogen (3.68%). The data show that the carbon content in ECE is significantly lower than that of activated carbon (93.50%) and graphite (98.30%). For HAE, the contents of C, H, O, and N are 48.69%, 4.72%, 35.20%, and 5.70%, respectively. Compared with the oxygen content (32.98%) of HA, the higher content of O in HAE indicates that HAE possesses more oxygen-containing groups.

**Table 4.** Element composition of carbonaceous matters (value in wt.%).

| Component | C | H | O | N | Sum |
|---|---|---|---|---|---|
| ECE | 75.50 | 2.85 | 4.33 | 3.68 | 86.36 |
| Activated carbon | 93.50 | 1.10 | 3.45 | – | 98.05 |
| Graphite | 98.30 | 0.03 | 0.01 | – | 98.34 |
| HAE | 48.69 | 4.72 | 35.20 | 5.70 | 94.31 |
| HA | 50.37 | 4.25 | 32.98 | 5.54 | 93.14 |

It is generally considered that the atomic ratios of C/N, H/C, and O/C can determine, describe, and analyze the structural features of humic acid. The value of C/N is often regarded as an essential parameter for the origin of HA substances [8]. The C/N ratio in Table 5 illustrates that the formation of HAE extracted from carbonaceous gold concentrate may be related to the nonvascular aquatic plants during the long-term geological evolution [16]. Further, the higher the value of H/C, the lower the degree of humification/aromaticity and the more the aliphatic structures in humic substances [17,18]. The H/C ratio of HAE is slightly higher than HA, indicating that the humification degree of HAE is significantly lower than that of HA [12]. This suggests the presence of relatively fewer aromatic materials and a large number of aliphatic structures in HAE. Unlike C/N and H/C, the O/C ratio mainly reflects the number of oxygenated groups in the HA substances [19,20]. The O/C value in Table 5 shows that HAE is higher than HA, indicating that the HAE contains more oxygenated groups than HA. It is well consistent with the elemental analysis.

**Table 5.** Atomic ratios of HAE and humic acid (HA).

| Atomic Ratio | C/N | H/C | O/C |
|---|---|---|---|
| HAE | 9.95 | 1.16 | 0.54 |
| HA | 10.61 | 1.01 | 0.49 |

3.2.2. SEM Analysis

In order to study the surface morphology of ECE and HAE separated from carbonaceous gold concentrate, SEM analysis was carried out. Figure 2 shows the SEM/EDS images of ECE. Most of the ECE particles are irregular blocks and the particle length is between 5.5 and 42 μm. Importantly, the surface of ECE particles exhibit a porous structure, which is likely to provide effective adsorption sites for the adsorption behavior [21]. SEM/EDS images of the HAE extracted from gold concentrate are presented in Figure 3 showing that most particles exist in the form of spheres. HAE consists of tiny particles with a particle size ranging from 40 to 400 nm. In addition, HAE exhibits intermolecular aggregation behavior, which is resulted from the presence of intermolecular forces between HAE molecules. Moreover, cohesive performance between HAE particles also confirms that the number of oxygen-containing structure in HAE is abundant [16]. It is in good agreement with the results of atom ratios analysis.

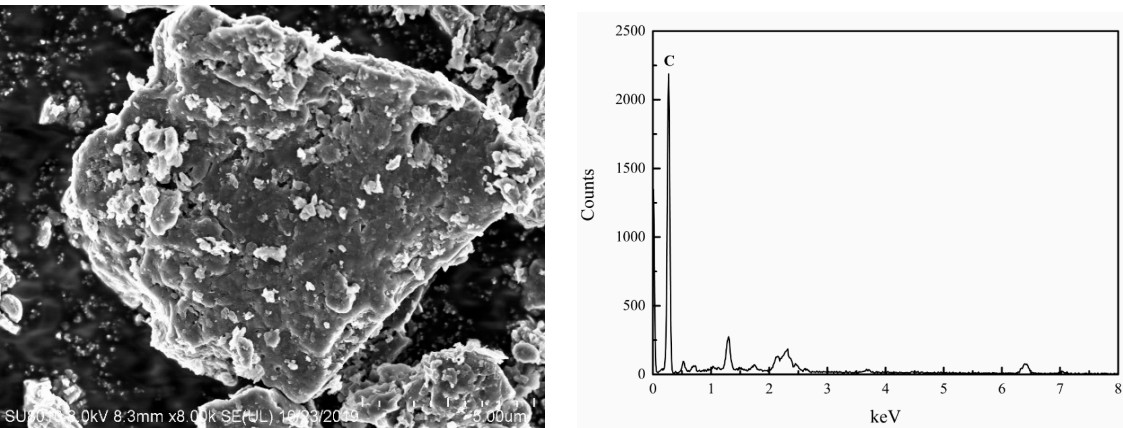

**Figure 2.** The SEM/EDS images of elemental carbon extract (ECE).

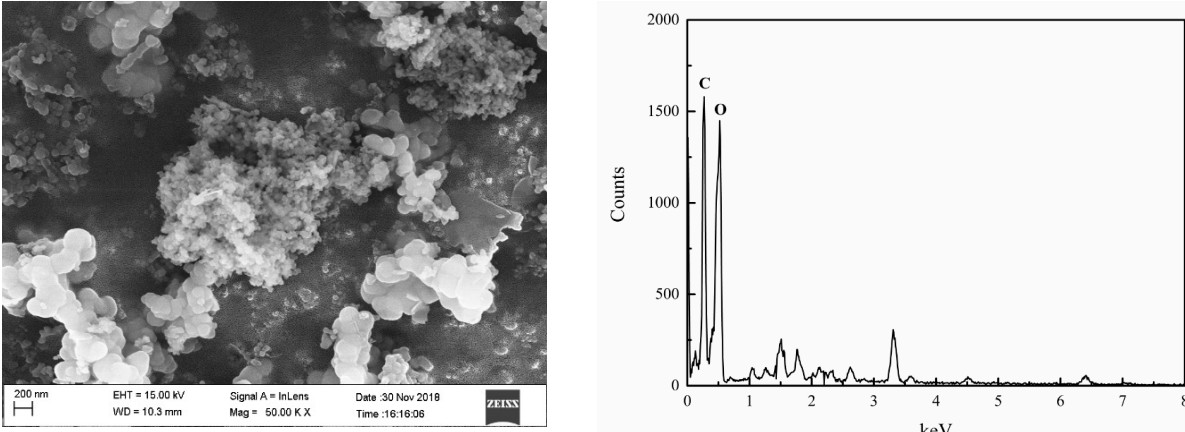

**Figure 3.** The SEM/EDS images of humic acid extract (HAE).

### 3.2.3. BET Area and Pore Analysis

The BET area and pore size of ECE, activated carbon, and graphite were measured. The detailed results are shown in Table 6. $S_{BET}$ of ECE is 20.35 $m^2 \cdot g^{-1}$, and the values of DA and VT of ECE are 31.42 nm and 0.16 $cm^3 \cdot g^{-1}$, respectively. The specific surface area of ECE is between graphite and activated carbon; it shows that the ECE has a significantly difference in structure compared with activated carbon and graphite, and this also indicates that the adsorption area of ECE is smaller than activated carbon but higher than graphite. According to the classification of pore size provided by IUPAC [22], the pore structure observed in SEM images (Figure 2) of ECE exists predominantly in the form of mesopores. Thus, the locations of these pores are the significant active adsorption sites where the adsorption reactions will most likely occur.

**Table 6.** The results of BET area and pore.

| Carbonaceous Matters | $S_{BET}$, $m^2 \cdot g^{-1}$ | $D_A$, nm | $V_T$, $cm^3 \cdot g^{-1}$ |
|---|---|---|---|
| ECE | 20.35 | 31.42 | 0.16 |
| Activated carbon | 745.60 | 25.45 | 1.89 |
| Graphite | 4.35 | 24.10 | 0.03 |
| HAE | 42.84 | 2.97 | 0.13 |
| HA | 27.45 | 3.13 | 0.09 |

BET, is an equation proposed by Brunauer, Emmet and Teller on the basis of the Lanmere equation to describe the theory of multi-molecular layer adsorption. $S_{BET}$, surface area; $D_A$, average pore diameter; $V_T$, total pore volume.

In accordance with the results of the pore analysis given in Table 6, it can be seen that DA and VT values of HAE are 2.97 nm and 0.13 cm$^3$·g$^{-1}$, respectively. Thus, based on the pore size classification of the adsorbent material provided by IUPAC, the pore diameter of HAE belongs to the mesoporous range. Based on the BET measurement, the surface area of HAE is 42.84 m$^2$·g$^{-1}$, which is significantly higher than HA. The relatively large specific surface area of HAE is related to the intermolecular agglomeration phenomenon observed in SEM images (Figure 3), which has been proved by the previous experimental study [23]. Furthermore, the large surface area of HAE also indicates that it may have sizeable adsorption area and more effective adsorption sites on the outer surface, which is beneficial to gold adsorption.

### 3.2.4. FTIR Analysis

The FTIR spectra of ECE, activated carbon, and graphite are presented in Figure 4. The peak of ECE and activated carbon at 3440 cm$^{-1}$, as well as the peak of graphite at 3450 cm$^{-1}$, are caused by the stretching vibration of O-H and NH groups [8,12]. For activated carbon, the absorption peaks at 2950, 2850 and 1390 cm$^{-1}$, respectively, are resulted in C-H bonds in CH$_3$ and CH$_2$ [24]. However, ECE shows a lower intensity vibration peak at 1384 cm$^{-1}$ due to the symmetrical bending vibration of the aliphatic CH$_3$ structure [25], indicating that ECE molecular arrangement contains a small amount of CH$_3$ structure. It is worth noting that ECE, activated carbon, and graphite all show an absorption peak at around 1630 cm$^{-1}$, these peaks are stemmed from the existence of the benzene ring [8]. This shows that ECE, activated carbon, and graphite all include a certain aromatic structure. The vibration peak of ECE at 1067 cm$^{-1}$ is associated with the C-O structure [26]. Activated carbon (1072 cm$^{-1}$) and graphite (1079 cm$^{-1}$) also show an adsorption peak with different intensity. In the region of 900–700 cm$^{-1}$, ECE, activated carbon, and graphite exhibit the adsorption peaks corresponding to the out-of-plane C-H of the aromatic hydrocarbon [24,27], which fully demonstrates that they all possess benzene ring or aromatic substances. Further, each of ECE and activated carbon contains an obvious adsorption peak within 600–500 cm$^{-1}$ (located, respectively, at 550 and 598 cm$^{-1}$), yet, graphite has no peak at the same wavelength. These peaks are attributed to the presence of halogenated hydrocarbons.

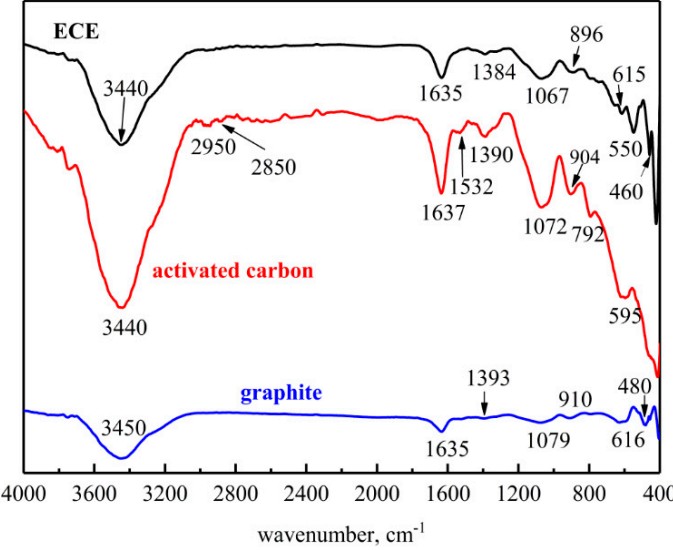

**Figure 4.** FTIR spectra of ECE, activated carbon, and graphite.

Figure 5 presents the FTIR spectra of HAE and HA. It shows that the functional groups compositions of HAE and HA are extremely similar and their chemical compositions are fairly complicated. Both HAE and HA contain a peak at 3430 cm$^{-1}$ corresponding to the stretching vibration of H-bonded OH and NH groups [28,29]. The occurrence of the

aliphatic chain is often revealed in 3000–2700 cm$^{-1}$ zone, the peaks at 2920 and 2850 cm$^{-1}$ of HAE and HA, respectively, are assigned as the asymmetric and symmetric stretching vibration of aliphatic C-H bonds in methyl and methylene groups [24,25]. In addition, the peaks appear at 1455 and 1385 cm$^{-1}$, which are resulted from the deformation vibration of methylene and the symmetrical bending vibration of methyl, respectively [16,30]. The presence of these peaks clearly revealed that HAE and HA contain a large amount of aliphatic structure. Then, the peak of HAE that appears at 1710 cm$^{-1}$ indicates the presence of carboxyl, carbonyl, aldehyde groups, or others [12,31,32]. However, the adsorption peak of HA at 1712 cm$^{-1}$ is relatively weak. Further, it can be seen that the peaks of HAE and HA at 1630 cm$^{-1}$ are regarded as a C=C bond in an aromatic ring. It is worth to note that some overlapping vibration peaks at 1630 cm$^{-1}$ cannot be excluded, such as C=O bond, hydrogen-bonding association, or an amide bond [12,20]. The peak occurring at 1120 cm$^{-1}$ attributes to the symmetric C-O stretching vibration of polysaccharides or polysaccharides-like substances [33,34]. It can be found that the peak intensity of HAE is higher than that of HA, indicating the enrichment of polysaccharides in HAE. The out-of-plane C-OH bending vibration related to carboxylic acid appears at 933 cm$^{-1}$ for HAE [24], but no absorption peak is observed in the spectra of HA. Similarly, the peak at 746 cm$^{-1}$ is only observed in the spectra of HAE, which explains the occurrence of out-of-plane C-H ortho-disubstituted of an aromatic ring in HAE [25]. In addition, a peak appears at 576 cm$^{-1}$ for HAE and at 620 cm$^{-1}$ for HA, which most likely arose from out-of-plane COO-wagging vibration of short-chain fatty acid [27].

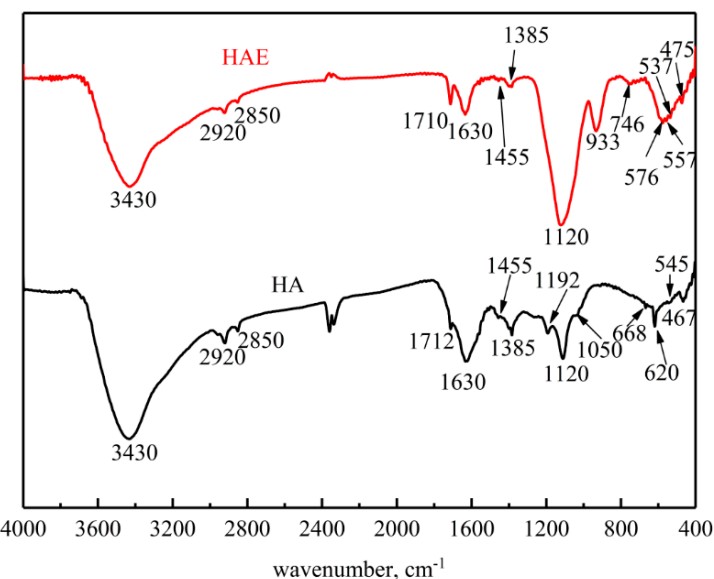

**Figure 5.** FTIR spectra of HAE and humic acid (HA).

### 3.2.5. UV–VIS Analysis of HAE

Humic substances in the UV region (200–400 nm) have considerable characteristics absorption [12,35]. According to Figure 6, the absorbance of HAE decreases sharply with the increase in wavelength, and the decrease in the UV region is more significant compared with the visible region. HAE possesses a UV characteristic absorption peak at 205 nm in the spectra, whereas a UV absorption peak appears at 206 nm for HA. Both peaks are possibly resulted from the structure of the benzene ring substituted by alkyl groups [36]. Thus, it is inferred that the benzene ring with alkyl groups may be involved in gold adsorption, but it still needs to be further confirmed by the experimental data. Furthermore, in this study, unlike HA, which has another UV peak at 238.5 nm, the absorption platform of HAE occurs in the wavelength range from 220 to 270 nm. This platform is attributed to the transition of π–π* electrons in phenolic aldehyde, polycyclic aromatic hydrocarbon structure, or aromatic compounds containing the double bonds (C=C, C=O, and N=N) [13,37,38]. The

results of UV–VIS spectra further support the existence of the benzene rings or aromatic structures containing phenolic aldehyde groups in the HAE, this is consistent with the structural analysis of FTIR spectra. It is assumed that these groups may perform an important role during gold adsorption.

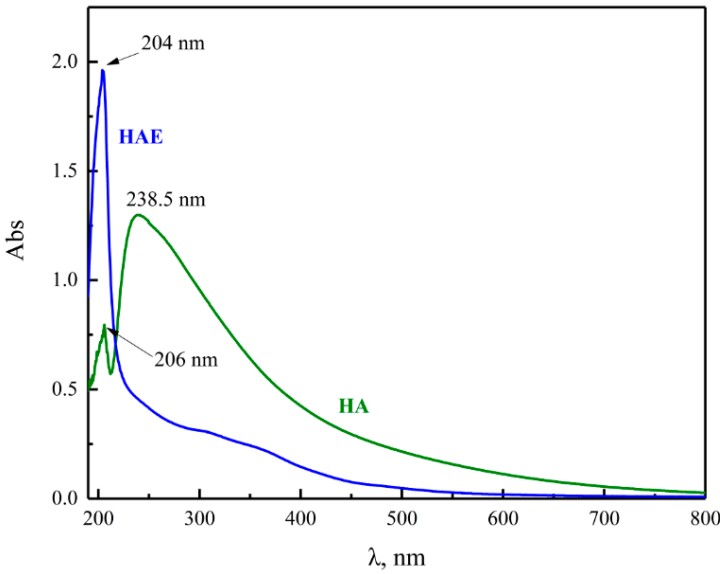

**Figure 6.** UV–VIS spectra of HAE and HA.

Previous studies have pointed out that the absorbance ratio of UV–VIS spectra at a specific wavelength can be used to indicate the characterization of the humic compound and provide valuable information on the chemical properties [36,39]. The characteristic absorbance ratios of HAE and HA are shown in Table 7. One of the most widely used parameters is $E_{465}/E_{665}$ (the ratio of intensity at 465 nm to that at 665 nm), which is expected to decrease with increasing degree of humification and average molecular weight [31,35,40]. The quotient $E_{465}/E_{665}$ of HAE is higher than HA, indicating that the average molecular weight and humification of HAE are lower than HA, and this also reveals that HAE contains more aliphatic groups compared with HA. The other vital indicators are $E_{270}/E_{400}$ and $E_{250}/E_{365}$, respectively. The ratio of $E_{270}/E_{400}$ is commonly associated with the chromophore content and the degradation degree of multiple functional groups to simple aromatic carboxylic products [39,41]. It is inferred by the $E_{270}/E_{400}$ value that HAE probably possess fewer chromophores and a relatively lower degree of degradation towards simpler aromatic carboxylic structures for comparison with HA. Further, the $E_{250}/E_{365}$ value has been reported to negatively correlate with the aromaticity of humic compounds [42]. The $E_{250}/E_{365}$ of HAE (2.08) is slightly higher than HA (1.93), implying that HAE has lower aromaticity. Therefore, the results of characteristic parameters are in agreement with the atom ratios analysis.

**Table 7.** Characteristics absorbance ratios of HAE and HA.

| Characteristic Absorbance Ratios | $E_{465}/E_{665}$ | $E_{270}/E_{400}$ | $E_{250}/E_{365}$ |
|---|---|---|---|
| HAE | 4.98 | 2.42 | 2.08 |
| HA | 3.35 | 2.94 | 1.93 |

### 3.2.6. XRD and Raman Spectra Analysis of ECE

The XRD analysis was carried out to analyze the phase composition of ECE, activated carbon, and graphite. The diffraction patterns are presented in Figure 7. It can be inferred that the (002) peak of ECE aligned well with activated carbon, and the shape of both peaks is extremely similar. Based on the (002) diffraction peaks, the graphitization (g) of

ECE, activated carbon, and graphite are calculated using formula (4) and (5), the g values are shown in Table 8. The g values for ECE, activated carbon, and graphite are 91.86%, 88.37%, and 95.35%, respectively. The data indicate that ECE has relatively low crystallinity compared with graphite, but higher than that of activated carbon. Moreover, the difference in the XRD pattern between ECE and graphite expresses that ECE has a relatively imperfect crystal structure and a certain degree of defects [43]. Thus, the structural composition of ECE is unique, i.e., it not only contains part of activated carbon but also has specific properties of graphite. More importantly, the (002) diffraction peak of carbonaceous matters is intimately associated with the gold-adsorbing capacity. The broad low graphite peak exhibits relatively high gold-adsorbing ability [43], while the sharp graphite peak shows that the carbonaceous matters have a relatively low gold-adsorbing capability [44]. Thus, it can be seen from the XRD patterns that the gold-adsorbing capability of graphite is the weakest, the gold-adsorbing capabilities of activated carbon and ECE are higher than that of graphite. However, it is unknown whether or not the (101) diffraction peak is associated with gold-adsorbing capacity of carbonaceous matters [44,45].

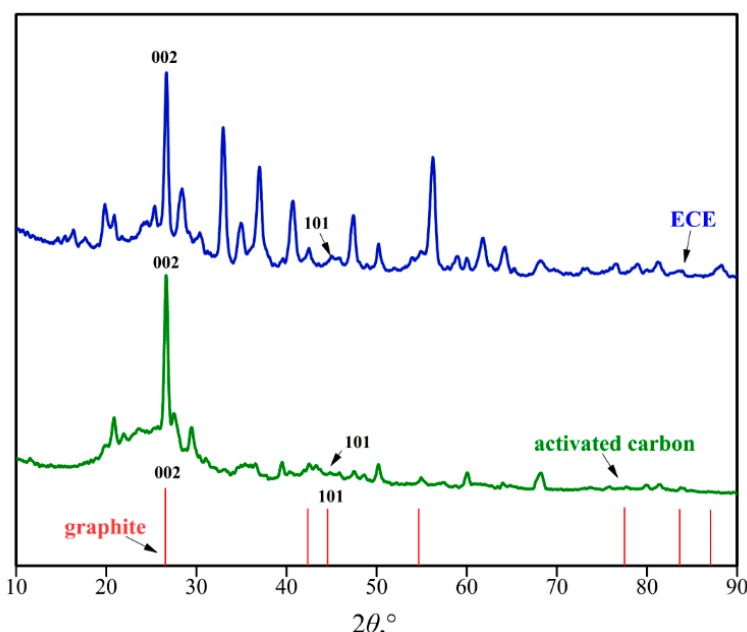

**Figure 7.** XRD diagram of ECE, activated carbon, and graphite.

**Table 8.** The g and R value of ECE, activated carbon, and graphite.

| Carbonaceous Matters | $d_{002}$ (nm) | g (%) | R |
|---|---|---|---|
| ECE | 0.3361 | 91.86 | 0.8123 |
| Activated carbon | 0.3364 | 88.37 | 0.9137 |
| Graphite | 0.3358 | 95.35 | 0.2904 |

The Raman spectra of ECE, activated carbon, and graphite are shown in Figure 8. All of them have D peaks (the defect and disorder of C atom crystal) and G peaks (the in-plane stretching vibration of C atom $sp^2$ hybrid) with different intensities in the vicinity of 1300 and 1580 cm$^{-1}$, respectively [46,47]. Related research proved that the D peak and G peak can reflect the gold-adsorbing capability of carbonaceous matters [48]. Activated carbon has the highest intensity of D peak and G peak, followed by ECE and graphite. It can be inferred that the adsorbing capability of ECE for gold is between activated carbon and graphite. In addition, R ratio (the ratio of the intensity of D peak to the intensity of G peak) can be used to describe and analyze the degree of defects in C atom crystal in carbonaceous matters. The higher the R ratio, the greater the degree of defects [49]. R

ratios of ECE, activated carbon, and graphite are 0.8123, 0.9137, and 0.2904, respectively (Table 8). Therefore, activated carbon has the most significant defect of C atom, whereas ECE is between activated carbon and graphite but close to activated carbon. This indicates that the crystal structure of ECE contains both activated carbon and the crystalline carbon structure same as graphite [50]. This finding is uniquely consistent with the result acquired from XRD analysis. Specifically, the R ratio is also associated with the gold-adsorbing capacity of carbonaceous matters. Previous study indicated that the higher the R-value, the lower the g value and the stronger the gold-adsorbing capability [50]. As a result, the decreasing order of the preferential adsorption of gold or gold complexions on carbonaceous matters is activated carbon, ECE, and graphite.

$$d_{002} = \lambda/2\sin\theta_{002} \tag{4}$$

$$g = (0.3440 - d_{002})/(0.3440 - 0.3354) \times 100\%, \tag{5}$$

where $\theta_{002}$ is the diffraction angle of (002) peak, $\lambda$ (nm) is the incident wavelength of Cu K$\alpha$ (0.15406 nm), $d_{002}$ (nm) is z-axis average layer spacing, and g (%) is the graphitization of carbonaceous matters.

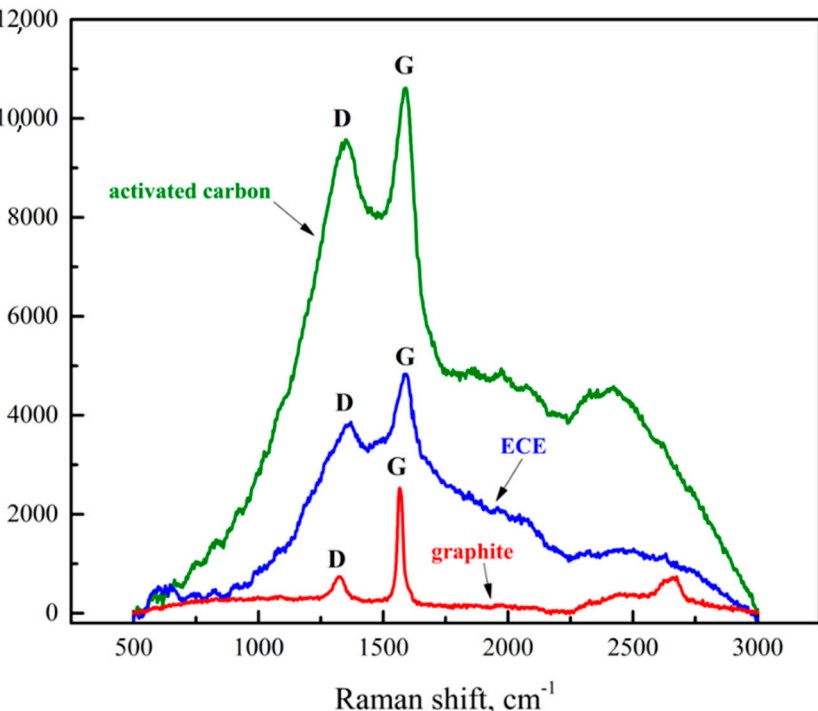

**Figure 8.** Raman spectra of ECE, activated carbon, and graphite.

*3.3. Gold-Adsorbing Experiments*

3.3.1. Gold-Adsorbing of ECE

The data of gold-adsorbing experiments of ECE separated from carbonaceous gold concentrate, activated carbon, and graphite are depicted in Figure 9. When the adsorption system is acidic, the amounts of gold adsorbed by ECE, activated carbon, and graphite are 470.46, 498.87, and 458.60 mg·g$^{-1}$, respectively. However, under alkaline conditions, the adsorption capacity of each of three is slightly reduced, the adsorbed amounts of ECE, activated carbon, and graphite are 449.02, 461.50, and 443.15 mg·g$^{-1}$, respectively. Whether it is in acidic or alkaline system, the gold-adsorbing capability of ECE is lower than activated carbon but higher than graphite.

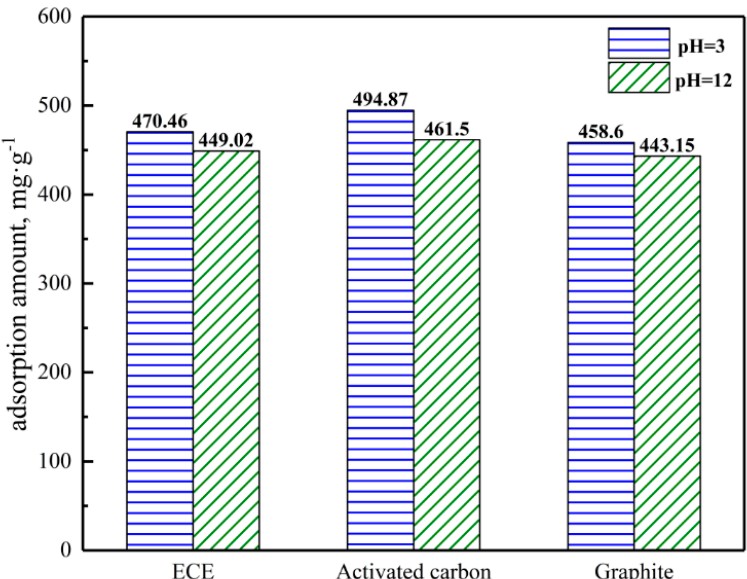

**Figure 9.** Gold-adsorbing experimental results of ECE, activated carbon, and graphite (pH 3 and 12, temperature 25 °C, adsorption time 5 h, and initial gold concentration 100 mg·L$^{-1}$).

### 3.3.2. Gold-Adsorbing of HAE

Gold-adsorbing experiments of HAE separated from carbonaceous gold concentrate and HA were carried out in this study. The results of gold-adsorbing experiments of HAE and HA are depicted in Figure 10. Under the acidic adsorption system, the gold adsorption amounts of HAE and HA are 357.60 and 393.80 mg·g$^{-1}$, respectively. As the humic acid dissolves in the alkaline solution, the amount of adsorbed gold decreases. The amount of gold adsorbed on HAE is 294.72 mg·g$^{-1}$ and on HA is 322.65 mg·g$^{-1}$. The difference in the amount of gold adsorbed on HAE and HA is cause by the minor differences in their molecular structures and chemical features.

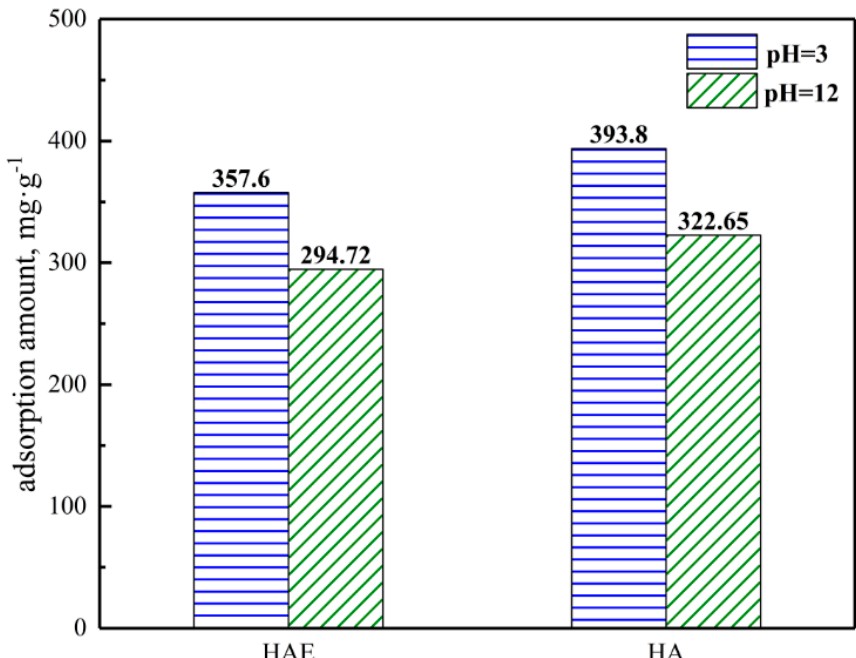

**Figure 10.** Gold-adsorbing experimental results of HAE and HA (pH 3 and 12, temperature 25 °C, adsorption time 5 h, and initial gold concentration 100 mg·L$^{-1}$).

### 3.3.3. Adsorption Process

SEM-EDS images, XRD, and FTIR spectra of ECE after adsorbing gold are listed in Figure 11. According to the SEM image, a lot of blocky particles are adsorbed on the surface of ECE. The EDS analysis shows that these particles are gold-containing matters, indicating that gold-adsorbing behavior mainly occurs at the mesoporous structure and outer surface of ECE. This is consistent with the conclusions from Sibrell and Miller that the gold adsorption process primarily occurred at pores, the defects, or edge areas of elemental carbon [51]. The result of XRD indicates that after the adsorption of gold, the crystal structure of ECE has been changed, e.g., the transformation of shape and intensity appear in the (101) diffraction peak; this also confirm that gold-adsorbing process would lead to the change in the structure of ECE. Furthermore, compared with Figure 4, there are minor differences between before adsorbing and after adsorbing in the spectra of FTIR. This also indicates that the adsorption of ECE to gold is mainly depended on the mesopore structure and active surface.

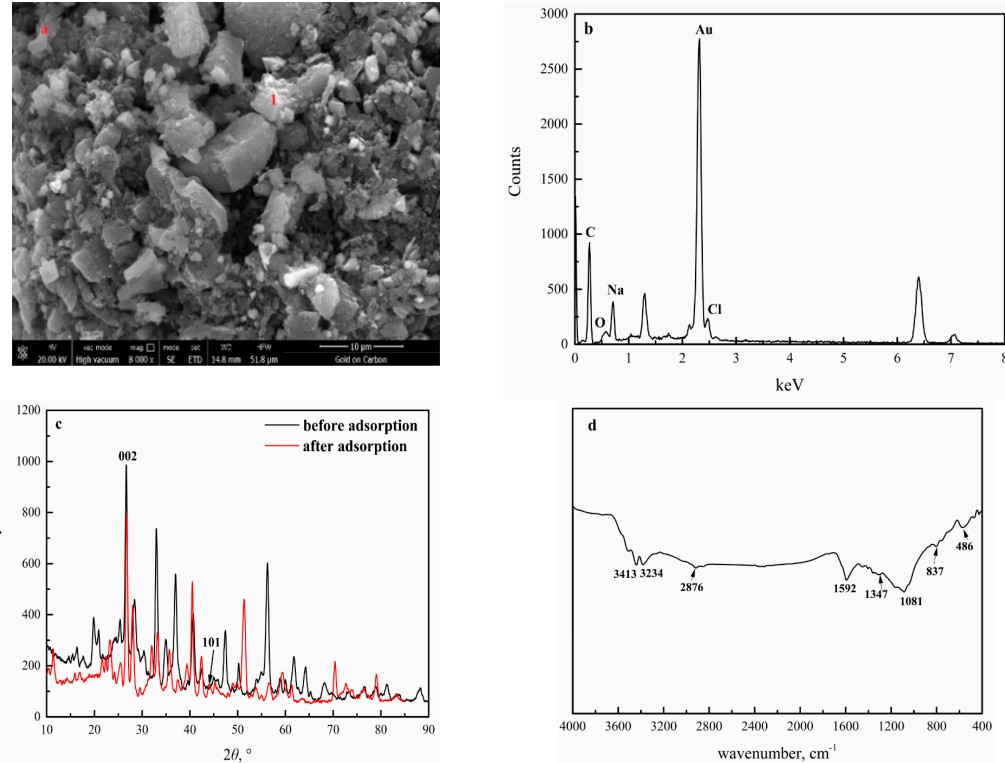

**Figure 11.** SEM (**a**), EDS (**b**), XRD, (**c**) and FTIR spectra (**d**) of ECE after adsorption (point 1 is the designated analysis area in the SEM image).

In Figure 12 are the SEM-EDS images, FTIR, and UV–VIS spectra of HAE after adsorption. A large amount of molecular agglomeration results in having higher surface activity and larger adsorption areas. In conjunction with the discovery of SEM-EDS images (Figure 12a,b) after adsorbing gold, the molecular aggregate structure of HAE is destroyed, and a large number of gold-containing particles are adsorbed on the outer surface of HAE. FTIR analysis shows that benzene ring and C-O bond of polysaccharide have changed after adsorption and that the various absorption peaks in the fingerprint region have also changed. Moreover, a red shift can be observed in the UV–VIS spectra, indicating that the spatial structure of HAE gets changed and the conjugation effect of HAE is enhanced. Importantly, the disappearance of the UV absorption platform indicates that the double bond structure (such as, C=C and N=N) is involved in the adsorption behavior. Therefore, gold can be bounded or ion-exchanged with certain functional groups to form new gold-

containing substances [52], the process of HAE adsorbing gold may be a combination of complexation, ion-exchange, and surface adsorption.

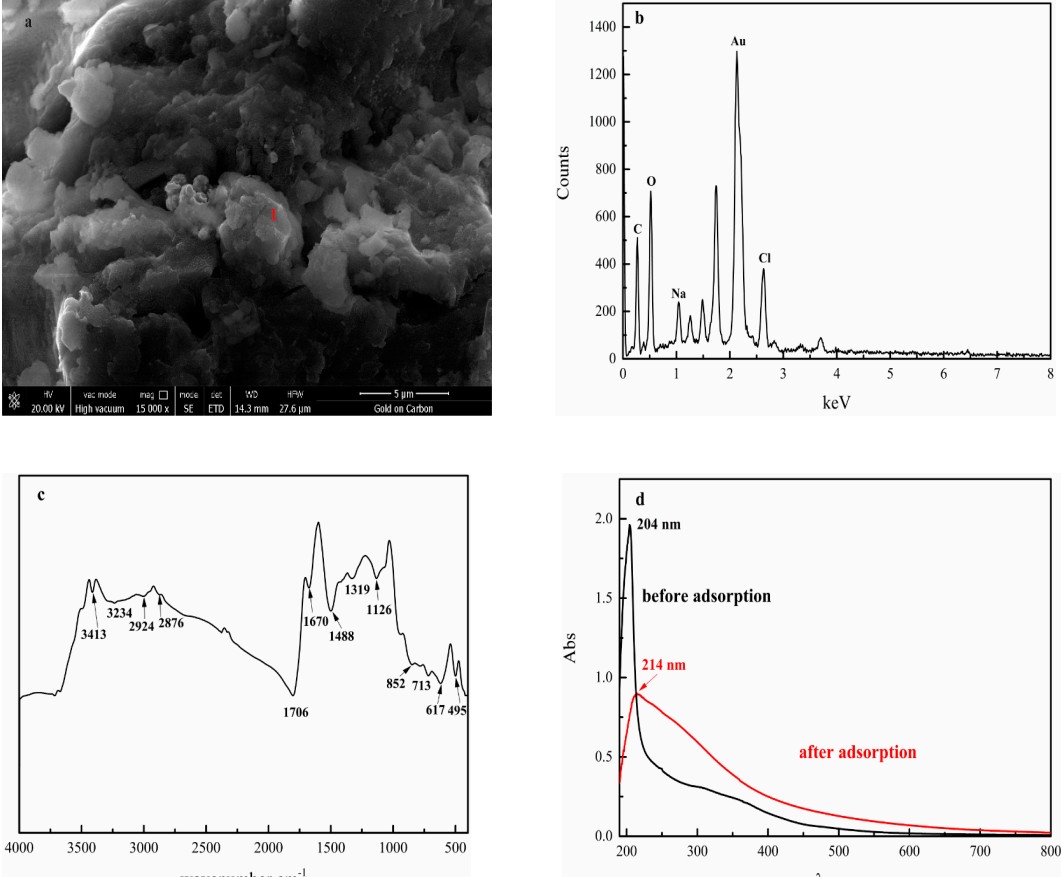

**Figure 12.** SEM (**a**), EDS (**b**), FTIR, (**c**) and UV–VIS spectra (**d**) of HAE after adsorption (point 1 is the designated analysis area in the SEM image).

Obviously, the effect on gold adsorption of ECE for gold is significantly higher than that of HAE, indicating that the adsorption efficiency of ECE occupies a dominant position between ECE and HAE. Herein, ECE is one of the carbonaceous matters that should be pretreated preferentially in processing the carbonaceous gold concentrate. Consequently, this study offers a new strategy to pretreat carbonaceous matters, which is based on the structural composition and the gold-adsorbing behavior of carbonaceous matters. For ECE, the optimal pretreatment method to effectively suppress or destroy the pore structure on the surface is utilizing technologies such as microbial oxidation, covering inhibition, or competitive adsorption. The way to reduce the adsorption capacity of HAE on gold is to destroy oxygenated functional groups and prevent the active sites from adsorbing the gold-containing compounds using technologies, e.g., photodegradation, biological oxidation, or chemical oxidation.

## 4. Conclusions

The structural characterization and gold-adsorbing experiments of ECE and HAE extracted from a carbonaceous gold concentrate were conducted employing a number of analytical methods. The following conclusions can be drawn.

(1) The concentrate sample used in this study contains 15.9 g·t$^{-1}$ Au, 1.70% inorganic carbon, and 4.36% organic carbon. The extracted amounts of ECE and HAE are 14.84–38.50 and 11.55–28.05 mg·g$^{-1}$, respectively, or $0.9 \times 10^3$–$2.33 \times 10^3$ g·t$^{-1}$ and $0.79 \times 10^3$–$1.69 \times 10^3$ g·t$^{-1}$, respectively.

(2) ECE particles mostly occur in irregular blocks with the size from 5.5 to 42 μm. Notably, the surface of ECE is mainly in the form of mesoporous with an average pore diameter of 31.42 nm. The specific surface area is 20.35 $m^2 \cdot g^{-1}$. The structural characteristic of ECE is similar to activated carbon, and the crystallinity is lower than graphite. R ratio of ECE is 0.8123, which is lower than activated carbon but much higher than graphite. In addition, the surface of ECE molecular structure possesses some aromatic, aliphatic, and oxygen-containing structure.

(3) HAE particles appear in spherical structures with a size mostly ranging mainly between 40 and 400 nm. HAE has high surface activity and effectively active adsorption sites associated with a large specific surface area of 42.84 $m^2 \cdot g^{-1}$. The average pore diameter is 2.97 nm belonging to mesopore. Importantly, HAE is a complex organic compound with various oxygenated functional groups, including carboxyl, hydroxyl, carbonyl, ethers, and others. The degree of humification of HAE is lower than HA based on the UV–VIS data, while the number of aliphatic structures in HAE is higher than that of HA.

(4) The gold adsorption amounts of ECE and HAE are 470.46 and 357.60 $mg \cdot g^{-1}$ under the acidic conditions, respectively. The adsorption amounts of ECE and HAE in an alkaline adsorption environment are 449.02 and 294.72 $mg \cdot g^{-1}$, respectively.

(5) SEM-EDS, XRD, and FTIR analysis can confirm that the adsorption process of ECE to gold mainly occurs on the surface and mesoporous structure, accompanying the change of the crystal and chemical structure. In terms of HAE, complexation of functional groups and the effective adsorption sites of the outer surface may be the primary approaches to the gold-adsorbing process.

**Author Contributions:** Writing—review and editing, H.N., H.Y., L.T.; funding acquisition, H.Y. All authors have read and agreed to the published version of the manuscript.

**Funding:** The work was supported by the Special Fund for the National Natural Science Foundation of China (U1608524), National Key R&D Program of China (2018YFC1902002) and the Zijin Mining Group Co., Ltd. (ZJKY2017(B)KFJJ01 and ZJKY2017(B)KFJJ02).

**Acknowledgments:** We greatly thank all anonymous reviewers for their constructive comments, which improved the manuscript.

**Conflicts of Interest:** The authors declare no conflict of interest.

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
