# Peer review of "Structural Characterization and Adsorption Capability of Carbonaceous Matters Extracted from Carbonaceous Gold Concentrate"

_minerals, doi:10.3390/min11010023_

Round 1

Reviewer 1 Report

Dear authors, I have read your work and want to focus on the following comments:

  1. Figures should be significantly improved because their quality is low, inscriptions are not easy to read, etc.
  2. Points 2.3-it is necessary to describe in more detail the methods: why all these operations are performed, reagents are added, etc. Alternatively, provide links to relevant techniques.
  3. Much attention is paid to analytical studies of the initial carbonaceous materials. However, there are no studies of the obtained products of the studied gold adsorption process using similar methods that are widely presented for the initial materials.
  4. The conclusions about the mechanisms of adsorption are based on assumptions that have no significant evidence.
  5. Cautious suggestions are made on the topic of the reduction of the sorption activity of the studied materials, but it would be better to support them by specific studies.
  6. In the conclusions, 3 out of 4 points are devoted to analytical studies of the initial carbonaceous materials, and only 1-to the actual adsorption process, and is also based on assumptions.

I suggest that the authors strengthen the evidence-based scientific part of their conclusions about the mechanisms of gold adsorption processes for the studied carbonaceous materials.

Author Response

Repely comments are uploaded in the form of Word file.

Reviewer 2 Report

Very nice paper with a very complete study, but that needs some modifications before it can be accepted.

Information missing

For a start, the Introduction needs much more details than what is provided.

Starting by the first sentence "Carbonaceous matters in carbonaceous gold ore are known to cause the “preg-robbing” phenomenon during the gold leaching".

Please explain, for readers not familiar with the subject, what is carbonaceous gold ore and from where it comes from, also what is the “preg-robbing” phenomenon and why and how it occurs, and provide a further explanation for the gold leaching (why, how and when).

Section 3.2.3 only shows the BET area and pore analysis in form of a Table (6). It would be much more useful to see also the N2 adsorption desorption isotherms and pore size distributions. Also to classify the type of isotherms, hystheresis, etc.

Figures

Some figures also need improvement, namely Figure 1. The resolution is low and the letter size is too small. It is very hard to read. 2 teta seems like 20.

Figures 2 and 3 - it would be useful to see some EDS images to accompany the SEM micrographs, if possible, in order to confirm the composition of the material.

Figures 4-6, 8 have low resolution that should be improved.

Figure 7 also has low resolution and small letter size, but also needs more changes. There are several peaks that are not identified, neither the phases nor the Miller indices. Why is the spectra of graphite not shown and only some lines are present?

Mystypes

line 147, correct carbonaceous (on the 3.2 section title)

line 149, "paramount component" sounds strange; "important component", or similar would be preferable.

Author Response

Dear Editors and Reviewers:

Thank you for your letter and for the reviewers’’ comments concerning our manuscript entitled “Structural Characterization and Adsorption Capability of Carbonaceous Matters Extracted from Carbonaceous Gold Concentrate” (ID:1026784). Those comments are all valuable and very helpful for revising and improving our paper, as well as the important guiding significance to our researches. We have studied comments carefully and have made correction which we hope meet with approval. Revised portion are marked in red in the paper. The main corrections in the paper and the responds to the reviewer’s comments are as following:

Responds to Reviewer 2 Comments:

Point 1: Figures should be significantly improved because their quality is low, inscriptions are not easy to read, etc.

Response 1: Thank you for your suggestions.  We have improved the quality of all Figures.

Point 2: Points 2.3-it is necessary to describe in more detail the methods: why all these operations are performed, reagents are added, etc. Alternatively, provide links to relevant techniques.

Response 2: Thank you for your comments. The extraction method of ECE is summarized according to literature review. The extraction of HAE is based on the method provided by the International Humic Substances Society (IHSS).

Point 3: Much attention is paid to analytical studies of the initial carbonaceous materials. However, there are no studies of the obtained products of the studied gold adsorption process using similar methods that are widely presented for the initial materials.

Response 3: Thank you for your comments. The carbonaceous matters in the carbonaceous gold ore have the “preg-robbing” activity. The structure of carbonaceous matters in different carbonaceous gold ore is different, and the physical and chemical properties are also different. Therefore, in this study, the gold adsorption experiments were conducted by the carbonaceous matter extracted from the carbonaceous gold ore without pre-treatment or improvement, testing their adsorption capacity for gold, comparing their adsorption amount of gold, clarifying which carbonaceous matters are dominant in the adsorption process.

Point 4: The conclusions about the mechanisms of adsorption are based on assumptions that have no significant evidence.

Response 4: Thank you for your suggestion. Indeed, the adsorption process is based on assumptions that have no significant evidence. Therefore, we conducted some tests on the carbonaceous matters after adsorbing gold to explain the adsorption process in Section 3.3.3.

Point 5: Cautious suggestions are made on the topic of the reduction of the sorption activity of the studied materials, but it would be better to support them by specific studies.

Response 5: Thank you for your comments. Indeed, it would be better to support reduction of the sorption activity by specific studies.  This is also the study direction we will focus on.

Point 6: In the conclusions, 3 out of 4 points are devoted to analytical studies of the initial carbonaceous materials, and only 1-to the actual adsorption process, and is also based on assumptions.

Response 6: Thank you for your comments. We have revised your comments in the conclusions.

Special thanks to you for your good comments.

Round 2

Reviewer 1 Report

Dear authors, I thank you for the work done and corrections made. i liked your revision and updated scientific information. I wish you success!